# Prenatal Diagnosis and Postnatal Outcomes of Left Brachiocephalic Vein Abnormalities: Systematic Review

**DOI:** 10.3390/jcm11071805

**Published:** 2022-03-24

**Authors:** Gerarda Gaeta, Vlasta Fesslova, Roberta Villanacci, Danila Morano, Massimo Candiani, Mirko Pozzoni, Margherita Papale, Silvia Lina Spinillo, Carmelina Chiarello, Paolo Ivo Cavoretto

**Affiliations:** 1Gynecology and Obstetrics Department, IRCCS San Raffaele Hospital, Via Olgettina 60, 20132 Milan, Italy; gerarda.gaeta14@gmail.com (G.G.); villanacci.roberta@hsr.it (R.V.); candiani.massimo@hsr.it (M.C.); pozzoni.mirko@hsr.it (M.P.); papale.margherita@hsr.it (M.P.); spinillo.silvia@hsr.it (S.L.S.); cavoretto.paolo@hsr.it (P.I.C.); 2Gynecology and Obstetrics Department, University Vita-Salute San Raffaele, Via Olgettina 60, 20132 Milan, Italy; 3Center of Fetal Cardiology, IRCCS Policlinico San Donato, Via Morandi 30, San Donato Milanese, 20097 Milan, Italy; moranodanila@gmail.com; 4Department of Congenital Cardiac Surgery, IRCCS Policlinico San Donato, Via Morandi 30, San Donato Milanese, 20097 Milan, Italy; carmelina.chiarello@grupposandonato.it

**Keywords:** congenital heart defect, fetal venous system, fetal echocardiography, fetal left brachiocephalic vein, fetal innominate vein, intrathymic left brachiocephalic vein, subaortic left brachiocephalic vein, retroesophageal left brachiocephalic vein, ultrasound

## Abstract

Abnormalities of the left brachiocephalic vein (LBCVA) are rare and poorly studied prenatally. An association with congenital heart defects (CHD), extracardiac and genetic abnormalities was described. The aim of our study was to estimate the rate and summarize the available evidence concerning prenatal diagnosis, associated anomalies, and outcomes of these anomalies. A systematic literature review was carried out selecting studies reporting on prenatal diagnosis of LBCVA, including unpublished cases from our experience. Frequencies were pooled from cohort studies to calculate prenatal incidence. Pooled proportions were obtained from all the studies including rates of associated CHD, extracardiac or genetic abnormalities and neonatal outcomes. The search resulted in the selection of 16 studies with 311 cases of LBCVA, with an incidence of 0.4% from six cohort studies. CHD occurred in 235/311 (75.6%) fetuses: 23 (7.4%) were major in cases of double, retroesophageal or subaortic course and 212 (68.2%) were minor in cases of absence (always associated with a persistent left superior vena cava) or intrathymic course. Data on other associated outcomes were scarce showing rare extracardiac anomalies (3.5%), rare genetic abnormalities (RASopathies and microdeletions associated with the retroesophageal course), and neonatal outcomes favorable in most cases, particularly in intrathymic forms.

## 1. Introduction

Mediastinal fetal venous system is poorly investigated in prenatal diagnosis. The International Society of Ultrasound in Obstetrics and Gynecology (ISUOG) recommends inclusion of the three vessels and trachea (3VT) view in the routine cardiac screening ultrasound [1,2]; 3VT makes it possible to perform an accurate examination of the upper mediastinum with the assessment of the fetal thymus and other vessels not usually included in a routine examination such as subclavian arteries and variants of the left brachiocephalic vein (LBCV). 

Normal LBCV originates from the joining of the left jugular and subclavian veins and crosses obliquely the upper mediastinum, anteriorly to the branches of the aortic arch and dorsally to the thymus, to join the vertical right brachiocephalic vein, thus originating the superior vena cava (SVC). 

The first case of an anomalous brachiocephalic vein was described by Kershner more than 100 years ago [3]. Nowadays, some LBCV anomalies (LBCVA) have been described in prenatal age, while a greater number has been reported postnatally, in surgical series, as an occasional finding during imaging or post-mortem. The cases described include absence or duplication, abnormal course and dimension. In particular, the subaortic and retroesophageal courses were found to be associated with intracardiac and genetic disorders, in contrast to the intrathymic course [4,5,6,7,8]. 

Since the evaluation of the LBCV is not part of the routine examination of fetal heart, its exact incidence and association with intracardiac, extracardiac, or genetic anomalies is still unknown, with reports ranging from 0.06% to 0.37% for the general population [9] and from 0.2 to 1.7% in a postnatal series of patients with congenital heart defects (CHD) [10,11,12]. 

In summary, little is known about clinical and prognostic significance of fetal LBCVA, and utility of routine examination of this vessel in prenatal diagnosis must be defined.

## 2. Objective

The aim of the present study was to summarize the available evidence concerning prenatal diagnosis and outcomes of LBCVA with particular interest to thr rate of associated cardiac heart defects (CHD), extracardiac anomalies (ECA), genetic abnormalities, and neonatal outcomes.

## 3. Materials and Methods

### 3.1. Search Strategy, Information Sources and Eligibility 

The MEDLINE, PubMed, Scopus, Google Scholar and EMBASE databases were searched using a combination of the following keywords: ((“left brachiocephalic vein” OR “innominate vein”) AND (“anomalies” OR “fetal” OR “prenatal” OR “intrauterine” OR “abnormal” OR “prenatal diagnosis” OR “dilatated” OR “dilation” OR “retroesophageal” OR “intrathymic” OR “ retrotracheal” OR “retroaortic” OR “subaortic” OR “double” OR “absent”)). This systematic literature review was conducted in accordance with the Preferred Reporting Items for Systematic Reviews and Meta-Analyses (PRISMA) statement (Figure 1) [13]. The EndNote software (available online: https://endnote.com, accessed on 30 August 2021) was used to remove duplicate articles.

Only studies written in English language available up to August 2021 and reporting prenatal diagnosis of abnormal LBCV were considered, including grey literature (two research congress abstracts) and six unpublished cases from our personal experience. Retrospective, longitudinal studies, case reports and case series were included. Pediatric, postnatal, and post-mortem series were excluded as they are based on a different population with a potential selection bias.

### 3.2. Study Selection, Data Collection, and Outcomes

The flowchart of study selection is shown in Figure 1 [13].

Data concerning neonatal outcomes of fetuses prenatally diagnosed with abnormal LBCV were collected and recorded in a dedicated database by two different authors (G.G. and M.P.). Data concerning neonatal outcomes of fetuses prenatally diagnosed with abnormal LBCV from studies clearly describing nine subgroups (intrathymic LBCV, subaortic LBCV, retroesophageal LBCV, abnormal course not specified, presence of two simultaneous anomalies of the LBCV, LBCV absent with a persistent left SVC, LBCV dilation, double LBCV, and a group of LBCV abnormalities in which the type of abnormality was not specified) were also considered and recorded separately, by two different authors. The outcomes analyzed were congenital heart defects (CHD), extracardiac structural abnormalities (ECA), genetic anomalies, termination of pregnancy (TOP), gestational age (GA) at prenatal diagnosis, maternal age at diagnosis, and pregnancy outcomes. Two authors (G.G. and M.P.) reviewed all the articles independently, and consensus was reached about relevance and inconsistencies. Any doubt and inconsistency were resolved by consulting senior authors (P.C., V.F.). The study did not require ethical approval. The component of the study including unpublished cases collected retrospectively from our center was based on a database where patients’ data were stored anonymized and complied with the ethical standards for human research established by the Declaration of Helsinki. 

### 3.3. Study Quality Assessment

Quality assessment of the included studies was achieved using the National Institute of Health (NIH) tool for the quality assessment of Case Series Studies (https://www.nhlbi.nih.gov/health-topics/study-quality-assessment-tools; accessed on 30 August 2021) (Table 1). This method was also recommended by the National Institute for Health and Care Excellence (NICE). The grades were attributed based on questions 1–9: “good” if questions 1, 6, and 7 (principal factors) were present; “fair” if two factors were present; and “poor” or “insufficient quality” if one factor was present. A global assessment (good, fair, and poor) according to the Agency for Healthcare Research and Quality (AHQR) was performed, the NICE and NIH standards were assigned to each study. For cohort studies, the Newcastle Ottawa scale was used [14] (Table 2). 

### 3.4. Local Cohort

After the year 2014 when we diagnosed the first case, all echocardiographies included routine clinical assessment of the left brachiocephalic vein with assessment of size, dilation, and course. All the abnormal cases were recorded in a prospective database managed at our centers. 

## 4. Results

Sixteen studies were retrieved from analysis of the literature with a total of 311 cases of LBCV anomalies (LBCVA), including six cases from our series (Appendix A, Table A1). The mean maternal age at diagnosis was 30.2 years, and the mean gestational age at diagnosis was 23.4 weeks. Fetal echocardiography was the diagnostic method of LBCV anomalies in 15 studies [7,8,15,16,17,18,19,20,21,22,23,24,25,26], and in our series, while fetal magnetic resonance was used in one study [27]. In studies in which the initial sample was reported, LBVCA appeared to have an incidence from 0.02% to 6.5% [7,16,18,23,27]. Only in one study it was possible to differentiate between the incidence of LBCVA in a population of fetuses with congenital heart disease and in normal fetuses, which was 0.5% and 2%, respectively [7]. Among the LBCV anomalies described, there were anomalies of the course (intrathymic, subaortic, retroesophageal course, and anomalies of the course not specified); number (absence or double LBCV); size (dilation); in one case (0.3%) of the 311 LBCVA, there were both abnormal intrathymic course and dilation [22], and there were also three cases (1%) out of the 311 with LBCV anomalies without specifying the type of anomaly but associated to anomalous supracardiac pulmonary venous returns [18] (Figure 2).

The most frequent LBCVA was absence of the LBCV associated with persistence of the left superior vena cava (LSVC) occurring in 177 (56.9%) cases of the 311 LBVCA; in five of these cases. concomitant absence of the superior right vena cava was also present [8]. Anomalies of the LBCV course were documented in 109/311 (35.1%) cases. Course anomalies were differentiated into intrathymic, subaortic, and retroesophageal, with rates of 14.8% (46/311), 9.7% (30/311), and 6.1% (19/311) of the total number of LBCVA (Figure 2) and rates of 42.2% (46/109), 27.5% (30/109), and 17.4% (19/109) of the total LBCV abnormalities, respectively. In addition, among the 109 anomalies of the course, other 14 cases were described; in these, however, the type of abnormal course was not specified, and all were associated with the persistence of the LSVC. The latter cases had a rate of 12.9% (14/109) on course anomalies and 4.5% (14/311) on total LBCVA. Dilation of the LBCV was reported in 6.1% (19/311) of the cases; a double LBCV was found in 0.6% (2/311) of the cases.

### 4.1. LBCVA and Congenital Heart Defects (CHD)

We considered as congenital heart defects (CHD) all structural intracardiac defects, anomalies of great vessels, or venous return (both systemic and pulmonary). The fetuses affected by CHD described in the group with various types of LBCVA were 235/311 (75.6%) (Table 3). In Figure 2, we showed the different types of CHD reported in the studies of our review and their associations with types of LBCV abnormalities. The relatively high percentage of CHD associated with LBCVA in our review is due to a large number of isolated LSVC that does not present major hemodynamic implications. This anomaly was present in 177 cases in which LBCV was absent [7,8,18,23], and also in 14 cases [18] in which the type of LBCV anomaly was not specified, with a total 191 cases of LSVC out of 311 LBCVA (61.4%). 

Considering only the major CHD (PAtr, TOF, AVSD with TA, and TAPVC), these occurred in 23 cases out of the 311 LBCVA (7.4%); all were associated with either double LBCV or retroesophageal and subaortic courses. 

Cao et al. [18] effectively described three cases of TAPVC and 14 cases of LSVC (above mentioned) in fetuses with LBCV anomalies but without specifying their type. An isolated persistent LSVC in association with absent LBCV should be considered a minor cardiac anomaly without major hemodynamic implications. Indeed, we are aware that LSVC is otherwise associated with more complex CHD or heterotaxies [28]. In total, there were 212 cases (68.2%) of minor CHD. 

There were 5/311 (1.6%) cases of right aortic arch and aberrant left subclavian artery (ALSA) and one (0.3%) case of atrioventricular septal defect (AVSD) with truncus arteriosus (TA) and one (0.3%) case of aberrant right subclavian artery (ARSA) (Appendix A, Table A1, Figure 2).

As for the association of specific courses with CHD, they were present in 13/19 (68.4%) cases of the retroesophageal LBCV course; in the subaortic LBCV course, CHD were described in 18 cases out of 30 (60%). In intrathymic course, CHD were reported in one case out of 46 (2.2%) (only a small VSD) [22].

Of the 19 cases of LBCV dilation described, six were associated with TAPVC, where the dilation of the LBCV was a consequence of increased venous return in this vessel as described by Sinkovskaya et al. [8]. The two cases of double LBCV were associated with TOF and RAA with ALSA and left DA. 

### 4.2. LBCVA and Extracardiac Anomalies (ECA)

In the cohort of fetuses with LBCVA, 11/311 cases (3.5%) of ECA were reported. In particular, there were five cases of intracranial venous and arteriovenous malformations (AVM) associated to LBCV dilation, four with a vein of Galen aneurysm, one with arteriovenous malformation of the dural sinus [8]. In cases with the retroesophageal course of the LBCV, there was one case of hydrops fetalis and VSD [19]; one case of esophageal atresia with tracheoesophageal fistula (EA-TEF); two cases of cystic hygroma and bilateral jugular lymphatic sac (JLS) [20], one case of single umbilical artery (SUA) [16], and one case with neurodevelopmental problems [29]. No cases of extracardiac anomalies associated with the intrathymic course of the LBCV were described in the papers reviewed; however, in our series, we observed one case with macrocephalus and facial dysmorphisms associated with the intrathymic course of the LBCV. 

### 4.3. LBCVA and Genetic Anomalies 

Screening or diagnostic genetic tests for chromosomal abnormalities with a combined test, cell-free fetal DNA (cffDNA) or amniocentesis, were reported only in 20 fetuses (20/311, 6.4%): of the 19 screening tests, 14 were negative and five were positive; eight cases underwent a further investigation with a diagnostic test by amniocentesis (one of these cases—without a screening test). Seven cases were reported to have a genetic anomaly, including three RASopathies, one 22q11 microdeletion syndrome, one trisomy 21, one case of 8p inverted duplication and deletion, and one case with 16p12.2 microdeletion (Appendix A). In all the three cases of RASopathy, there was the retroesophageal course of the LBCV, as well as in the case of 22q11 microdeletion syndrome, of 8p inverted duplication and deletion, and 16p12.2 microdeletion; trisomy 21 was associated with the absence of the LBCV and a bilateral SVC. 

### 4.4. Neonatal Outcomes 

Neonatal outcomes were available in 68 fetuses (Appendix A). Fifty-four had a normal outcome (54/68; 79.4%), while in three cases of neonates delivered at full term, an associated ECA was detected (one case of EA-TEF, one case with neurodevelopmental problems, and one case of our series had macrocephalus and facial dysmorphism as described above). Two cases of preterm birth were reported (one < 34 weeks and one < 37 weeks of gestation); termination of pregnancy (TOP) was chosen in nine cases (9/67, 13.4%) with associated major anomalies. 

### 4.5. Local Cohort

Our local cohort was based on a retrospective analysis of a prospectively collected database of a tertiary Fetal Echocardiography Center at San Donato Hospital and Fetal Medicine Center of San Raffaele Hospital, Milan, starting from December 2014 (when the first case was diagnosed) to March 2021. During this period, a total of 2388 fetal echocardiographies were performed, and our six cases represent 0.3% of the population examined, and they were all with intrathymic courses. The mean maternal age at diagnosis was 32.2 years, and the mean gestational age at diagnosis was 26.3 weeks of gestation. Of these six cases, four had an isolated form, one case presented a muscular VSD closed soon after birth, and the other case mentioned above had macrocephalus and facial dysmorphisms, without a specific syndromic diagnosis at the postnatal genetic test. Analysis of the karyotype was performed by amniocentesis only in one pregnancy with LBCVA isolated, and the karyotype was normal. Cell-free fetal DNA testing was performed in two pregnancies and was normal in both. In the remaining three cases, a first trimester combined screening test was performed, with no evidence of anomalies and low risk for common trisomies. All the cases were delivered at full term, with normal neonatal and maternal outcomes. 

## 5. Discussion

### 5.1. Principal Findings

This review found a prenatal incidence of LBCV anomalies of about 0.4% and showed an abnormal course of the LBCV that occurred in 35.1% of the cases (retroesophageal, subaortic, intrathymic, and other course anomalies not specified), absence of the LBCV with a left SVC—in 56.9%, dilation—in 6.1%, duplication—in 0.6%. In one case, there were either course (intrathymic) and caliber (dilation) abnormalities of the LBCV (1/311 (0.3%)), and in three cases out of the 311 LBCVA (1%), the type of LBCV abnormality was not specified. LBCVA are rarely associated with extracardiac defects (3.5%), whereas CHD occurred in 68% and 60% of the cases of retroesophageal and subaortic course, respectively, and in about 2% of the intrathymic forms. In 212 of the 311 (68.2%) LBCVA cases, minor CHD were present, especially a persistent LSVC associated with absent LBCV, which represented a finding of no serious clinical importance in the majority of cases. Instead, major CHD were described only in 23 (7.4%) fetuses with LBCVA; all were associated with either double LBCV or retroesophageal and subaortic courses. Therefore, in about two thirds of the cases of LBCVA, minor CHD were present, while only in one third of the cases there were major CHD. There are few genetic data reported in the reviewed papers, and genetic anomalies were mainly associated with the retroesophageal course. Neonatal outcomes were also scarcely reported, but in most cases, especially in the presence of the intrathymic course and absence of the LBCV, they were optimal. It is important to note that of the five cases of AVM associated with LBCV dilatation, four were vein of Galen aneurysms, one—with an arteriovenous malformation of the dural sinus. 

### 5.2. Interpretation

The exact prevalence of this venous anomaly in the general population is rare and not well-defined yet, with different rates and estimates resulting from different studies. In particular, the left brachiocephalic vein is not routinely checked during prenatal ultrasound examinations; therefore, it is not possible to establish exactly the prevalence of LBCV anomalies in the prenatal period. The literature on prenatal diagnosis reports abnormalities of the course, size, and number of LBCV (Figure 3 and Figure 4). From the data reported in this review, some types of LBCV abnormalities, such as the subaortic, retroesophageal course, dilation, and absence are apparently more associated with some specific cardiac abnormalities. The subaortic and retroesophageal courses seem to be more associated with conotruncal anomalies and RAA; this is in accordance with the postnatal data present in the literature [5,12,30]. Absence of the LBCV was associated with the persistence of the LSVC in all the cases described in our review. A persistent LSVC is usually associated with specific cardiac anomalies or heterotaxies [28], while in the reports of our review, it was apparently associated with the absence of the LBCV. LBCV dilation seems to be more associated with supracardiac TAPVC in prenatal diagnosis as already reported in the literature on postnatal data [31,32]. Sinkovskaya et al. established reference values in prenatal diagnosis for the assessment of the diameter of the LBCV, asserting that the normal size of this vessel increases during pregnancy from values ranging from 0.7 mm at 11 weeks to 4.9 mm at term of pregnancy [8]. Only two cases of double LBCV have been described prenatally, both associated with other cardiac abnormalities (one case with TOF and the other with RAA, ALSA, and left DA), but the sample is too small to draw conclusions. On the other hand, in the papers reviewed, the intrathymic course of the LBCV was less associated with other cardiac abnormalities and was usually isolated [7]. A smaller number of extracardiac abnormalities associated with LBCVA has been described; among those, the association between intracranial vascular malformations and LBCV dilation, such as the vein of Galen aneurysm, is interesting [8]. Limited data are available concerning the genetic abnormalities associated with LBCVA; of the six cases with genetic abnormalities, three had RASopathies, and all had the retroesophageal course of the LBCV. Neonatal outcomes were reported only for 63 fetuses with the intrathymic course, dilation or absence of the LBCV; 79.4% of them had normal outcomes.

### 5.3. Clinical Implications

Given the easy identification of the LBCV in the 3VT view of the upper mediastinum, its routine investigation in prenatal screening ultrasound could help to improve the detection of other CHD, particularly, in the case of the retroesophageal or subaortic course, known to be frequently associated with conotruncal anomalies and with RAA, that may sometimes cause obstructive respiratory problems, but this apparently did not occur in the papers reviewed. Furthermore, in the cases with dilation of the LBCV, there is a greater risk of abnormalities of the pulmonary venous return or arteriovenous malformations, but in these cases dilation of the right heart section will help to make a correct diagnosis of these situations. 

Furthermore, an early diagnosis of LBCVA is important to avoid complications from any postnatal left-sided vascular interventions [33,34]. The knowledge of an LBCV abnormality can be useful in some surgical procedures performed in the chest, such as pacemaker implantation [12]; in a cardiopulmonary bypass—for the establishment of adequate alternative venous drainage; for the cannulation of the superior vena cava that must be performed with extreme care to avoid obstruction of the anomalous vein [35].

Furthermore, the presence of an LBCV anomaly may complicate the surgical exposure of the pulmonary arteries in case of creation of the subclavian artery-to-pulmonary artery anastomoses, and an anomalous LBCV could obscure the surgical field in the creation of a subclavian-to-pulmonary arterial shunt [11,36] and in the ligation of a patent duct [6,11,35]. Lastly, the LBCV can be mistaken for a pulmonary artery during the reconstruction of the right ventricular outlet and angioplasty of bilateral central pulmonary arteries [12].

### 5.4. Strengths and Limitations

The major strength of this study is reporting of a comprehensive and up-to-date summary of cases described in the literature concerning diagnosis of different types of LBCV anomalies in the prenatal period with correlation to other concomitant cardiac and extracardiac malformations, genetic abnormalities, and neonatal outcomes. From the data exposed in our study, the finding of certain types of LBCVA provides different diagnostic insights and can be helpful in prenatal counseling.

Limitations of the study are related to heterogeneity of the design of the included studies and to the fact that, due to the rarity of some subtypes of these anomalies, it was difficult to estimate their incidence in the general population. Data on the genetic surveys and newborn outcomes reported are only few; therefore, this does not allow providing generalization and causal associations with this anomaly. 

### 5.5. Future Research

Prospective studies should be carried out to further investigate the incidence of the various types of left brachiocephalic vein anomalies and their association with genetic abnormalities and postnatal outcomes, only scarcely investigated up to now.

The diagnosis of such an anomalous vessel during prenatal scanning could be easily obtained by trained operators obtaining an extra view very close to that obtained to assess the three vessels and trachea view with no extra cost or effort. We should consider this anomaly as a potential marker of cardiac, extracardiac, and/or genetic abnormalities.

## 6. Conclusions

This is the first review study investigating the different types of LBCV abnormalities in the prenatal period, assessing their association with cardiac and extracardiac anomalies, genetic abnormalities, and neonatal outcomes. The data reported show that the anomaly of this vessel is a rare phenomenon, with an estimated rate of about four out of 1000 fetuses.

Subaortic, retroesophageal, dilated, and absent LBCV appeared potentially associated to cardiac and genetic abnormalities, while the intrathymic form seems to be only an isolated variant of the systemic venous return, without clinical implications. 

## Figures and Tables

**Figure 1 jcm-11-01805-f001:**
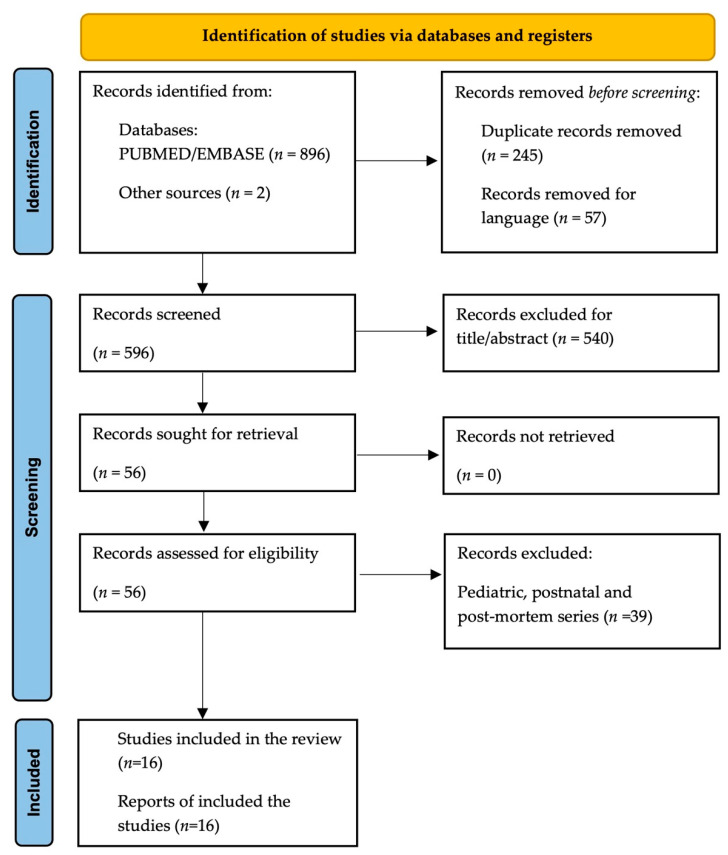
PRISMA flowchart of study selection.

**Figure 2 jcm-11-01805-f002:**
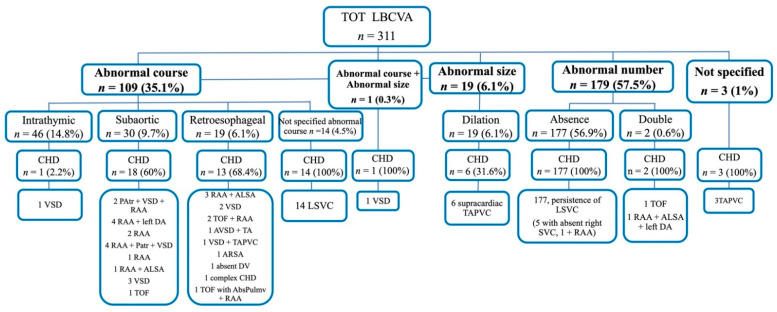
Number and percentage (%) of LBCVA (left brachiocephalic vein anomalies) and CHD (congenital heart defects) associated. LBCVA = left brachiocephalic vein anomalies; CHD = congenital heart defects; VSD = ventricular septal defect; RAA = right aortic arch; PAtr = pulmonary atresia; DA = ductus arteriosus; ALSA = aberrant left subclavian artery; TOF = tetralogy of Fallot; AVSD = atrioventricular septal defect; TA = truncus arteriosus; AbsPulmv = absent pulmonary valve; TAPVC = total anomalous pulmonary venous connection; ARSA = aberrant right subclavian artery; DV = ductus venosus; LSVC = persistent left superior vena cava; SVC = superior vena cava.

**Figure 3 jcm-11-01805-f003:**
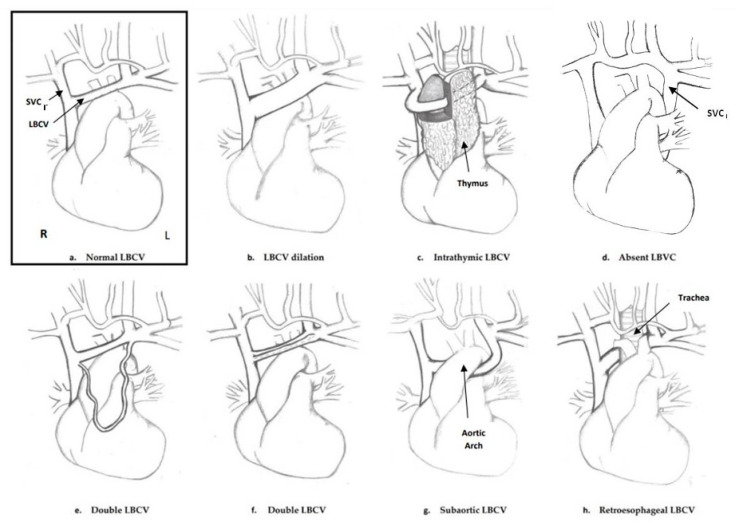
Illustration of the normal left brachiocephalic vein (LBCV) anatomy and LBCV anomalies: (**a**) normal LBCV; (**b**) dilated LBCV; (**c**) intrathymic LBCV; (**d**) absent LBCV with a persistent left superior vena cava (LSVC); (**e**,**f**) double LBCV; (**g**) subaortic LBCV; (**h**) retroesophageal LBCV. SVCr = right superior; SVC_l_ = persistent left superior vena cava.

**Figure 4 jcm-11-01805-f004:**
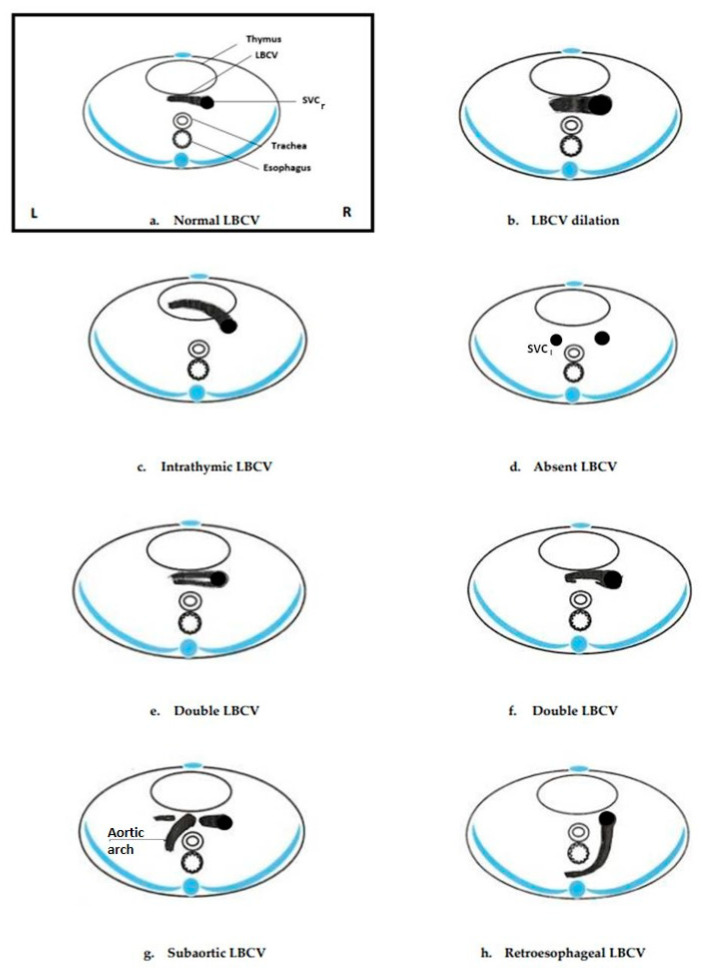
Illustration of transverse sections of the upper mediastinum as seen on an ultrasound scan with the left brachiocephalic vein (LBCV) recognizable: (**a**) normal LBCV; (**b**) LBCV dilated; (**c**) intrathymic LBCV; (**d**) absent LBCV with LSVC; (**e**,**f**) double LBCV; (**g**) subaortic LBCV; (**h**) retroesophageal LB. SVCr = right superior vena cava; SVC_l_ = persistent left superior vena cava; L = fetal left side; R = fetal right side.

**Table 1 jcm-11-01805-t001:** Quality assessment of case series studies.

Author, Year	1. Was the Study Question or Objective Clearly Stated?	2. Was the Study Population Clearly and Fully Described, Including a Case Definition?	3. Were the Cases Consecutive?	4. Were the Subjects Comparable?	5. Was the Intervention Clearly Described?	6. Were the Outcome Measures Clearly Defined, Valid, Reliable, and Implemented consistently Across all Study participants?	7. Was the Length of Follow-up Adequate?	8. Were the Statistical Methods Well-Described?	9. Were the Results Well-Described?	GRADE AHRQ Standards
Sinkovskaya et al. 2012	Yes	Yes	Yes	Yes	Yes	Yes	Yes	Yes	Yes	**Good**
Gilboa et al. 2013	Yes	Yes	NR	CD	Yes	Yes	Yes	NA	Yes	**Good**
Cheng et al. 2017	Yes	Yes	NA	CD	Yes	Yes	No	NA	Yes	**Fair**
Corbacioglu et al. 2017	Yes	No	NA	NA	Yes	Yes	No	NA	Yes	**Fair**
Chan et al. 2019	Yes	No	NR	CD	Yes	No	No	NA	No	**Poor**
Ma et al. 2020	Yes	No	NA	NA	Yes	No	No	NA	No	**Poor**
Rakha et al. 2021	Yes	Yes	NA	NA	Yes	Yes	Yes	NA	Yes	**Good**
Karmegaraj et al. 2021	Yes	Yes	NA	NA	Yes	No	Yes	NA	Yes	**Fair**
Mori et al. 2021	Yes	Yes	NA	NA	Yes	No	No	NA	Yes	**Poor**
Zhao et al. 2021	Yes	Yes	NA	NA	Yes	Yes	Yes	NA	Yes	**Good**
Linnane et al. 2021	Yes	Yes	NA	NA	Yes	Yes	Yes	NA	Yes	**Good**

AHRQ = Agency for Healthcare Research and Quality.

**Table 2 jcm-11-01805-t002:** Newcastle Ottawa scale. Quality assessment of cohort studies using the Newcastle Ottawa scale.

Title, Year	Selection	Comparability	Outcome	Total
Karl, 2016	★★★★	★★	★★★	★★★★★★★★★
Shah, 2018	★★★★	★★	★★	★★★★★★★★
Dong, 2018	★★★	★	★★★	★★★★★★★
Cao, 2019	★★★★	★	★★	★★★★★★★

**Table 3 jcm-11-01805-t003:** Summary of proportions of prenatal diagnosis of left brachiocephalic vein anomalies (LBCVA).

Author, Year	Total Number of the Fetuses Screened for LBCVA	Total Number of LBCVA	Number of LBCVA Isolated	Number of LBCVA Non-Isolated	Number of LBCVA with CHD	Number of LBCVA with Any ECA
Sinkovskaya E., 2012	NR	91	6	85	80 (68 LSVC)	5
Gilboa Y., 2013	NR	8	8	0	0	0
Karl K., 2016	2962	37	25	12	12 (4 LSVC)	0
Cheng Y.K.Y., 2017	NR	2	0	2	2	1
Corbacioglu A., 2017	NR	1	0	1	1	0
Shah N., 2018	1540	9	7	2	2 (2 LSVC)	0
Dong S.Z., 2018	7282	9	1	8	8	0
Chan Y.M., 2019	NR	4	0	4	2	2
Cao H., 2019	2011	131	11	120	120 (117 LSVC)	0
Ma B., 2020	NR	1	1	0	0	0
Han J., 2020	31,356	7	1	6	5	1
Rakha S., 2021	NR	1	0	1	1	0
Karmegaraj B., 2021	NR	1	1	0	0	0
Mori M., 2021	NR	1	1	0	0	0
Zhao Y., 2021	NR	1	1	0	0	0
Linnane N., 2021	NR	1	0	1	1	1
Current series	2388	6	4	2	1	1
TOTAL n (%)	43,539	311	67 (21.5)	244 (78.5)	235 (75.6) CHD of which 191 (61.4) LSVC	11 (3.5)

LBCVA = left brachiocephalic vein anomalies; NR = not reported; CHD = congenital heart disease; LSVC = left superior vena cava; ECA = extracardiac anomalies.

## Data Availability

Not applicable.

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
