# Peer review of "Prenatal Diagnosis and Postnatal Outcomes of Left Brachiocephalic Vein Abnormalities: Systematic Review"

_jcm, 2022, doi:10.3390/jcm11071805_

Round 1

Reviewer 1 Report

The present paper is well written and addresses an actual topic in fetal medicine.

There are few papers on the abnormalities of the left brachiocephalic vein(LBCVA) and therefore a systematic review of the literature in valuable.  Authors conducted their search in a very well organized way and raised adequate conculsions. Even though novelty of the study can be considered average, overall merit is high. I have no concerns about the quality of the manuscript. 

Author Response

Dear reviewer,

thank you for reviewing our manuscript and for your kind appreciation. We send you the new version of the manuscript. 

Reviewer 2 Report

Thanks for the opportunity to review this paper. 

In this article the authors present a systematic review of antenatally diagnosed left brachicephalic vein anomalies, with types, associated features and clinical outcomes. This paper will be useful for clinicians performing antenatal scans, counselling parents or looking after infants with these congenital abnormalities. However, in the current format it is difficult to find the clinically relevant information from the manuscript and better organisation of the paper could make it much more readible and digestible for clinicians. 

Specific comments:

Line 120: I would suggest that the author use  capital letters for AND and OR in this complex medline search and also use parentheses as appropriate

Line 162: "The most frequent LBCVA was absence associated with the persistence of the left superior vena cava (LSVC), anomaly present in 177 (56.9%) cases of 311 LBVCA" - There is confusion whether persistent left superior vena cava is part of the defect or an associated anomaly (the authors discuss it both ways). I think it is better discussed as part of the defect. 

Line 168: "42.8%, 27.2% and 17.3% on total LBCV abnormal course respectively" I think this should add up to 100% percent, but it does not. 

Table 2; This table is very difficult to oversee in the current format, I think it would be much better in a landscape format in the final paper, as currently the column headers are just too long and are broken in multiple lines (sometimes within words), making it very difficult to read.

Table 2 legend: The paper uses a large number of abbreviations, some of them are given here, but they are also extensively used in the main text. I think the best would be to put them at the beginning or the end of the paper in one place as list of abbreviations.

4.1. LBCVA and intracardiac anomalies (ICAs) - This section contains important information about the risk of associated anomalies but it is very difficult to read or get the information from it. Using a table instead would make it much clearer in my opinion. 

Line 190: Here persistent left superior vena cava is discussed again, now as ICA (intra-cardiac anomaly), which it clearly is not. Admittedly, it belongs to "congenital heart disease" as it is recognised and managed by cardiologists and heart surgeons, similarly to TAPVD. But discussing it in both places makes it confusing. 

Table 3: Again, I think is this table left persistent SVC is included among "intracardiac anomalies"?

Table 4: On the other hand, this Table is not very useful, most rows (representing papers) are comopletely NR ("not reported"), it could be better discribed as text, with mentioning the most papers have not reported genetic associations.

Line 269 Please see my comments above about left persistent SVC

5.3 Clinical complications: I think in some cases these malformations presented as "vascular rings" or "slings" causing neonatal symptoms due to obstruction of oesophagus and/or the trachea. However, this is not mentioned or discussed in the paper at all. 

Appendix A: This graph is more informative and understandable than all the other tables and should belong to the main paper.

Appendix C: This Figure is also very useful. If it is original artwork of the authors it should come to the main paper. If they have re-used an already published figure, reference should be provided. 

Minor comment: the manuscript would benefit from language edits.

Reviewer 3 Report

The paper entitled “Prenatal diagnosis and postnatal outcome of left brachiocephalic vein abnormalities: Systematic Review” by Gaeta and colleagues is a review article about abnormalities of the left brachiocephalic vein (LBCV) detected prenatally. The aim of the study, which is based on a systematic review of the literature and on 6 cases reported by the authors, is to estimate the rate and to summarize the available evidences concerning prenatal diagnosis and outcome of this anomaly with particular interest to the rate of associated intracardiac and extracardiac anomalies or genetic abnormalities. The authors conclude that abnormalities of LBCV is a rare phenomenon and some types appear associated with cardiac and genetic abnormalities such as subaortic, retroesophageal, dilated and absent LBCV, while intrathymic form seems to be only an isolated variant of the systemic venous return, without clinical implications.

The article is well written, interesting and the prenatally detection of the malformation may have helpful clinical implications. However, the authors might consider the following comments:

  • Line 194. Specify if the ALSA was in right aortic arch.
  • Line 196. Specify if the ARSA was in left aortic arch?
  • Line 207 and 208. Was TAPVC associated with LBVC or the vertical vein was connected with the LBCV determining its dilation?
  • Telling that the prenatal diagnosis of a dilated LBVC could help to increase the detection of intracranial venous and arteriovenous malformations is quite misleading. It would be better to add the association of the dilation of the right atrium and of the right ventricle.
  • In the Table 1 is not possible to read the line between “Author year” and “GRADE AHRQ standards”.
  • The Table 2 is quite confusing and is not easy to read.
  • The authors should consider revision of the paper by a native English speaker, as there are some grammatical errors. Please check through carefully.

Round 2

Reviewer 2 Report

Thanks for your response. I feel that the authors have sufficiently addressed all my points and the paper has improved a lot.